# Hepatic LKB1 Reduces the Progression of Non-Alcoholic Fatty Liver Disease via Genomic Androgen Receptor Signaling

**DOI:** 10.3390/ijms22157904

**Published:** 2021-07-23

**Authors:** Jun H. Heo, Sang R. Lee, Seong Lae Jo, Je-Won Ko, Hyo-Jung Kwon, Eui-Ju Hong

**Affiliations:** 1Department of Veterinary Biochemistry, College of Veterinary Medicine, Chungnam National University, Daejeon 34134, Korea; heojh94@o.cnu.ac.kr (J.H.H.); srlee5@cnu.ac.kr (S.R.L.); jsr7093@o.cnu.ac.kr (S.L.J.); 2Department of Veterinary Toxicology, College of Veterinary Medicine, Chungnam National University, Daejeon 34134, Korea; rheoda@cnu.ac.kr; 3Department of Veterinary Pathology, College of Veterinary Medicine, Chungnam National University, Daejeon 34134, Korea; hyojung@cnu.ac.kr

**Keywords:** testosterone, LKB1, NAFLD, AMPK

## Abstract

The incidence of non-alcoholic fatty liver disease (NAFLD) increases in males aged >45 years, which indicates that androgens are associated with the development and/or progression of NAFLD, although excess dietary intake is the primary causative factor. However, it is uncertain how androgens are involved in the metabolic process of NAFLD, which is associated with the state of steatosis in hepatocytes. To investigate whether androgen receptor (AR) signaling influences NAFLD development, the state of steatosis was monitored in mouse livers and hepatocytes with or without androgens. As a result, hepatic lipid droplets, expression of AR, and phosphorylation of AMP-activated protein kinase (AMPK) and acetyl-CoA carboxylase (ACC) increased in the presence of testosterone. Concurrently, the expression of LKB1, an upstream regulator of AMPK, was increased by testosterone treatment. We observed that the fluctuation of AMPK-ACC signaling, which plays an important role in lipogenesis, depends on the presence of testosterone and AR. Additionally, we demonstrated that testosterone bound AR was recruited to the promoter of the *LKB1* gene and induced LKB1 expression. Our study highlights a novel mechanism by which testosterone modulates NAFLD development by inducing the mRNA expression of *LKB1*.

## 1. Introduction

Non-alcoholic fatty liver disease (NAFLD) refers to the state of steatosis, which is characterized by the excessive accumulation of triglycerides (TGs) in hepatocytes, regardless of alcohol assumption [1,2]. NAFLD is distinguished from non-alcoholic steatohepatitis (NASH) by the presence of inflammation. NASH, which is accompanied by inflammation, is a more advanced form than NAFLD and can progress to cirrhosis and liver cancer as hepatocarcinoma [3,4]. Thus, NAFLD must be managed to prevent progression to NASH or beyond. Obesity is a representative risk factor related to the incidence of NAFLD [5]. This is attributed to the promotion of hepatic de novo lipogenesis (DNL), when the excess energy intake, composed of carbohydrates and/or fat, is met [6]. Acetyl-CoA carboxylase (ACC) is the key enzyme of the rate-limiting step since it converts acetyl-CoA to malonyl-CoA, known as an initiator of lipogenesis. AMP-activated protein kinase (AMPK) is an upstream regulator that functions as a sensor of the energy status. AMPK senses the AMP/ATP ratio in the body and regulates the phosphorylation of ACC [7]. In other words, the liver stores excess energy in the form of TGs into hepatocytes via AMPK-ACC signaling.

Additionally, sex differences have been mentioned as another risk factor. Statistical studies have consistently revealed that the incidence of NAFLD is higher in males than in females [8]. In males, the incidence increases around 45 years of age. Meanwhile, the levels of androgens begin to decline naturally [9]. This suggests that the presence and/or activation of androgens may be associated with NAFLD development. Sex steroid androgens participate in the regulation of energy metabolic homeostases, such as obesity and glucose intolerance, and are associated with lipid metabolism [10,11]. Androgens induce an increase in the expression and/or phosphorylation of AMPK, leading to the inactivation of APMK-ACC signaling [12]. However, the exact mechanism by which androgens modulate AMPK-ACC signaling is still unclear. Therefore, we have revealed the detailed action of the mechanism between androgens and AMPK-ACC signaling and whether this action is linked to NAFLD development.

Next, we focused on liver kinase B1 (LKB1), which is an upstream regulator of AMPK [13]. LKB1, encoded by the gene, interacts with testosterone as well as AMPK [14]. In particular, the mRNA expression of *LKB1* is decreased in adipocytes, depending on the concentration of the testosterone, which stimulates AMPK phosphorylation [15]. Interestingly, unlike adipocytes, we observed that the mRNA expression of *LKB1* increased in the presence of testosterone in hepatocytes. Concurrently, a decrease in AMPK-ACC signaling and lipid accumulation was observed. This evidence suggests that testosterone, one of the androgens, suppresses NAFLD development as it plays an important role in the induction of transcription and/or translation of *LKB1*.

## 2. Results

### 2.1. Androgens Suppressed Hepatic TG Accumulation Involving in Inflammation and Fibrosis

To determine whether hepatic TG accumulation depends on the levels of androgens, male mice were subjected to surgical castration (orchidectomy; ODX) and were injected with 2.5 mg/mL testosterone (T) every 3 days. In addition, we divided the mice into the normal diet group (ND) and the high-fat diet group (HD) to confirm whether androgens participate in lipid metabolism (Figure 1A). Hematoxylin and eosin (H&E) staining of their livers revealed that the accumulation of lipid droplets was higher in the HD group than in the ND group. In particular, the accumulation of droplets significantly increased in the HD-ODX group compared to the HD-Naïve and HD-ODXT groups in which endogenous or exogenous androgens were sufficient (Figure 1B). Similarly, when we quantified the area of the lipid droplets stained with Oil Red O, a defined increase was observed in the HD group compared to the ND group. The quantified level of lipid accumulation in the HD-ODX group appeared to be more progressive than the HD-Naïve group (1.21 fold, *p* < 0.05) and the HD-ODXT group (3.48 fold, *p* < 0.05) (Figure 1B,C). In accordance with the ingested diet, the bodyweight of HD groups was higher than that of ND groups (*p* < 0.05) like the increased hepatic TG accumulation (Figure 1D).

Furthermore, to assess whether this liver steatosis state has progressed to a more serious stage, i.e., NASH or cirrhosis, we performed several additional analyses to assess liver damage. Serum levels of alanine aminotransferase (ALT), a general marker of hepatocellular injury [16], were significantly increased in HD-ODX mice, when compared to HD-Naïve (2.18 fold, *p* < 0.05) and HD-ODXT (2.28 fold, *p* < 0.05) mice (Figure 2A). The mRNA levels of interleukin-alpha (IL-α), an inflammatory factor, was increased in the ODX groups, compared to the Naïve groups, ND-Naïve vs. ND-ODX (1.44 fold, *p* < 0.05) and HD-Naïve vs. HD-ODX (1.69 fold, *p* < 0.05) (Figure 2B). Similarly, the mRNA levels of IL-β were also increased in the ODX groups, when compared to the Naïve groups, ND-Naïve vs. ND-ODX (1.98 fold, *p* < 0.05) and HD-Naïve vs. HD-ODX (2.50 fold, *p* < 0.05). Furthermore, the IL-β level of HD-ODXT decreased compared to HD-ODX (49%, *p* < 0.05) (Figure 2B). 

Representing the induction of fibrotic responses in the liver [17], TGF-β mRNA levels increased in HD-ODX mice, as compared to HD-Naïve mice (2.08 fold, *p* < 0.05) and HD-ODXT mice (1.89 fold, *p* < 0.05) (Figure 2C). When we performed Masson’s trichrome staining to detect fibrotic areas, the level of fibrosis was significantly higher in the HF-ODX group than in the HD-Naïve (1.52 fold, *p* < 0.05) and HD-ODXT (3.00 fold, *p* < 0.05) groups, similar to the TGF-β mRNA levels (Figure 2D). These results indicate that androgens may induce the development of NAFLD and facilitate the progression of NAFLD to NASH or cirrhosis.

### 2.2. Testosteronereduces Lipid Synthesis by Modulating AMPK-ACC Signaling via LKB1

The developmental mechanism of NAFLD is attributed to DNL [6]. DNL involves the regulation of AMP-activated protein kinase (AMPK) and acetyl CoA carboxylase (ACC), namely AMPK-ACC signaling [7]. Therefore, we investigated whether androgens modulate AMPK-ACC signaling. We then performed Western blotting since the activation of AMKP-ACC signaling is modulated by the phosphorylations of AMPK and ACC. In the presence of androgens, the ratios of phosphorylated AMPK (pAMPK) and total AMPK (AMPK) and pACC and ACC decreased in ND-ODX mice compared to those in ND-Naïve mice (pAMPK/AMPK; 53%, *p* < 0.05) (pACC/ACC; 6%, *p* < 0.05) (Figure 3A). The pAMPK/AMPK ratio was significantly decreased in HD-ODX mice than that in HD-ODXT mice (94%, *p* < 0.05). However, the significance of pACC/ACC was not clear (Figure 3A). These effects were also observed in an in vitro experiment using SNU-423 cells, in which the androgen receptor (AR) is expressed at higher levels among hepatocyte cell lines. Strikingly, when the cells were treated with 10 nM testosterone, the relative level of pACC was increased compared to the DMSO-treated vehicle (1.37 fold, *p* < 0.05) (Figure 3B). Additionally, we arranged the enriched lipid condition by treating cells with fatty acids such as palmitate (330 µM) and oleate (670 µM). Following fatty acid treatment, the ratio of pAMPK and pACC increased in the cells treated with the testosterone compared to the cells treated with DMSO-treated vehicle (pAMPK/AMPK; 1.15 fold, *p* < 0.05) (pACC/ACC; 1.15 fold, *p* < 0.05) (Figure 3B). To further examine that testosterone modulates the AMPK-ACC signaling, we performed the *AR* overexpression into Hep3B cells. The relative level of pAMPK increased in the cells treated with 10 nM testosterone compared to the DMSO-treated vehicle (1.38 fold, *p* < 0.05) (Figure 3C). In the *AR* overexpressed cells, the ratio of both pAMPK and pACC increased in the cells treated with 10 nM testosterone compared to the DMSO-treated vehicle (1.21 fold, *p* < 0.05) (Figure 3C). This intimates that testosterone, one of the androgens, suppresses AMPK-ACC signaling via phosphorylation and leads to the development of NAFLD.

Next, we focused on the LKB1, known as the primary upstream regulating factor of AMPK [18]. A recent study using adipocytes showed that LKB1 responds to sex steroid hormones, and the hormones interact with the receptor bound to them (e.g., androgen and estrogen receptors) [15]. Then, we devoted to whether the argument applies to hepatocytes. First, we performed an LKB1 knockdown assay was conducted using the SNU-423 cell line, to examine whether LKB1 participates in the upstream regulation of AMPK, leading to the phosphorylation of ACC. LKB1 knockdown was effective in the presence of fatty acids (49%, *p* < 0.05) (Figure 4A). Expectedly, AMPK phosphorylation decreased with the down-regulation of LKB1 expression (28%, *p* < 0.05) (Figure 4A).Concurrently, ACC phosphorylation also decreased when LKB1 was knocked down (69%, *p* < 0.05) (Figure 4A). To verify if testosterone stimulates a series of fluctuations from LKB1 expression to phosphorylation of AMPK-ACC, testosterone supplement assay depending on time point (0 h, 3 h, 6 h, 12 h, 24 h, 48 h) was conducted with SNU-423 cell line. Interestingly, the result showed alteration in the expression and phosphorylation had a regular flow (Figure 4B). This suggests that LKB1 is operated as an upstream regulating factor of AMPK and the expression stimulated by testosterone, in hepatocytes as well.

### 2.3. Testosterone Modulatethe Expression Level of LKB1 by Binding to AR

The amount and activity of AR have to be observed as androgen’s physiological effects are mediated by AR signaling on most occasions [19,20,21]. In an in vivo study, the mRNA expression of the AR increased in the presence of androgens. Although no significant difference was observed in ND-Naïve vs. ND-ODX mice, a meaningful decrease was observed in HD-ODX mice compared to that in HD-Naïve mice (83%, *p* < 0.05) and HD-ODXT mice (71%, *p* < 0.05) (Figure 5A). Similar to that of AR, the level of LKB1 prominently decreased in HD-ODX mice, when compared to that in HD-Naïve mice (63%, *p* < 0.05) and HD-ODXT mice (55%, *p* < 0.05) (Figure 5A). 

Similarly, mRNA and protein expression levels of the AR and LKB1 increased when testosterone was added to SNU-423 cells, compared to vehicle alone-treated cells vs. testosterone alone-treated cells (*AR*; 1.37 fold, *p* < 0.05) (*LKB1*; 1.63 fold, *p* < 0.05) (AR; 3.36 fold, *p* < 0.05) (LKB1; 1.82 fold, *p* < 0.05) (Figure 5B,C). To further verify whether the expression of LKB1 depends on testosterone and AR, we performed an additional experiment using the AR antagonist bicalutamide and *AR* overexpression assay. When bicalutamide (5 µM) was added to the SNU-423 cells, unexpectedly, the mRNA levels of *AR* were increased, rather than suppressed, compared to the testosterone alone-treated cells vs. the vehicle-treated cells in the presence of bicalutamide (1.34 fold, *p* < 0.05) and in the testosterone alone-treated cells vs. the testosterone-treated cells in the presence of bicalutamide (1.75 fold, *p* < 0.05) (Figure 5B). Nevertheless, the protein levels of AR significantly decreased in the testosterone alone-treated cells vs. the vehicle-treated cells in the presence of bicalutamide (43%, *p* < 0.05) and in the testosterone alone-treated cells vs. the testosterone-treated cells in the presence of bicalutamide (73%, *p* < 0.05) (Figure 5C). Along with the decreased AR protein level, LKB1 was also decreased at both the mRNA and protein levels in the testosterone-alone treated cells vs. the vehicle-treated cells in the presence of bicalutamide (*LKB1*; 41%, *p* < 0.05) (LKB1; 64%, *p* < 0.05) and in the testosterone alone-treated cells vs. the testosterone-treated cells in the presence of bicalutamide (*LKB1*; 78%, *p* < 0.05) (LKB1; 89%, *p* < 0.05) (Figure 5B,C). Then, an *AR* overexpression assay was conducted to further corroborate the interaction between the AR and LKB1 using the Hep3B cell line with low *AR* expression. When testosterone was administered, no significant fluctuation in the mRNA and protein levels of AR and LKB1 was observed (Figure 5D,E). Meanwhile, after the *AR* overexpression assay, the levels of AR and LKB1 changed in the presence of testosterone. The expression levels of AR and LKB1 increased at both the mRNA and protein levels compared to the vehicle after overexpression vs. testosterone-treated after overexpression (*AR*; 3.08 fold, *p* < 0.05) (*LKB1*; 1.29 fold, *p* < 0.05) (AR; 1.87 fold, *p* < 0.05) (LKB1; 1.27 fold, *p* < 0.05) (Figure 5D,E). This evidence suggests that testosterone and AR participate in regulating LKB1 expression. Furthermore, the possibility that AR, as a nuclear receptor, modulates the transcription of the *LKB1* gene directly was not ruled out.

As proposed in a previous study [15], we checked whether the testosterone and its AR are recruited to the promoter of the *LKB1* gene by performing a chromatin immunoprecipitation (ChIP) assay (Figure 6A). We found that AR is recruited to the promoter of the *LKB1* gene and the intensity was 3.91% compared to the ‘5% input’ sample (Figure 6B). Meanwhile, using the same chromatin of the ChIP sample, we additionally investigated whether the AR interacts with the *AMPK* promoter. However, the results showed no such interaction (Figure 6B). These findings support that the androgen and AR complex induces the expression of the *LKB1* gene, leading to increased LKB1 levels modulating AMPK-ACC signaling.

## 3. Discussion

The incidence of NAFLD has increased in modern society, with an increasing population of individuals with obesity [22]. This surge is more prevalent in males over 45 years of age than in females of the same age [8]. Considering the decline in the total androgen levels in males around the age [9], the relationship between androgens and NAFLD is an important topic, although it is established that NAFLD is strongly linked to obesity and type2 diabetes [23,24]. In line with this argument, the present study examined the effects of a fat-rich diet and showed that orchiectomized mice, deprived of endogenous androgens, are predisposed to liver steatosis compared to naive mice. This evidence suggests that the presence of androgens might protect against the development of NAFLD.

When mice were fed a fat-rich diet, body weight and hepatic TG accumulation concurrently increased than the mice fed with a normal diet. The increase of hepatic TG accumulation was prominent in orchiectomized mice, compared to naïve mice and the mice injected with testosterone after castration. Meanwhile, unlike the increased TG accumulation, body weight was lowest in the orchiectomized mice, among the mice fed with a fat-rich diet. It is considered that testosterone suppresses hepatic TG accumulation and increases the body weight due to an increase in fat-free mass [25]. Along with the degree of TG accumulation, orchiectomized mice were also more vulnerable to inflammation and fibrosis. This implies that androgens can prevent the development of NAFLD and progression from NAFLD to steatosis-induced NASH with fibrosis, under a fat-rich diet.

First, we observed the fluctuation of hepatic DNL, as a recent study showed that hepatic TG accumulation is induced via DNL in patients with NAFLD [26]. AMPK-ACC signaling plays a major role in hepatic DNL [27]. The phosphorylation of AMPK and ACC decreased in orchiectomized mice, leading to a reduction in hepatic TG accumulation. Likewise, we found that AMPK-ACC signaling was inactivated when testosterone was administered, particularly under the condition of fatty acid supplementation and *AR* overexpression in vitro assay. Concurrently, the mRNA levels of *AR* increased in vivo when androgens levels were sufficient. In addition, the mRNA levels of *AR* increased in testosterone-treated cells. The *AR* mRNA levels surged more sharply when cells were treated with bicalutamide, an AR antagonist. This increase is considered a compensatory response to the steep suppression of AR activation [28], as the AR antagonist and the media to eradicate androgens inside the cells were simultaneously used. Unlike mRNA expression, the antagonistic effect of bicalutamide affected protein expression. This suggests that testosterone which is one of the androgens, by cooperating with AR, modulates hepatic TG synthesis and accumulation via AMPK-ACC signaling.

To assess whether the complex bound with testosterone and AR (AR complex) is directly recruited to the promoter of the *AMPK* gene, we performed a ChIP assay using the SNU-423 cells treated with testosterone. However, the results of the ChIP assay did not show the recruitment of the AR complex to the promoter of the *AMPK* gene. This result suggests that testosterone does not directly modulate *AMPK* expression. Despite the evidence, we could not exclude the genomic response of testosterone since the increased AMPK phosphorylation paralleled activation of the AR. 

A recent study revealed that LKB1, an upstream regulator of AMPK, interacts with sex steroids in a genomic manner in adipocytes [15]. Therefore, we investigated whether LKB1 cooperates with androgens in the same way in hepatocytes. The results showed that *LKB1* mRNA expression increased when androgens and/or testosterone were present at sufficient levels both in vivo and in vitro, and in turn, the protein expression increased in vitro. Along with the increased LKB1 expression, the phosphorylation of AMPK and ACC was also raised. This implies that LKB1 mediates the signaling between androgens and AMPK phosphorylation in hepatocytes. Likewise, we speculated whether androgens drive *LKB1* expression in a genomic manner and performed a ChIP assay to assess the expression of the *LKB1* gene. The results showed that the AR complex was recruited to the promoter of the *LKB1* gene. As in previous ChIP studies on AR, the AR element sequence motif at the AR binding site is presently presumed to be “ACATTTGT” in part of the *LKB1* gene promoter [29].

In summary, our study suggests that androgens, particularly testosterone, can prevent hepatic TG accumulation by suppressing hepatic DNL. Androgens bound to AR modulate *LKB1* transcription through direct recruitment to the promoter of the *LKB1* gene. In turn, increased LKB1 expression modulates the phosphorylation of AMPK-ACC signaling, leading to the suppression of hepatic TG synthesis and accumulation. Therefore, these findings are meaningful because androgens deficiency might be another risk factor of NAFLD in males, and the control of hormone homeostasis prevents NAFLD from progressing to steatosis-induced NASH, cirrhosis, and hepatocellular carcinoma.

## 4. Materials and Methods

### 4.1. Antibody

Antibodies used in the present study were shown in Table 1 below.

### 4.2. Animals

C57BL/6N WT male mice were accommodated and experimented in the pathogen-free facility at Chungnam National University in line with the Chungnam National University Animal Care Committee (CNU-00606). We provided them with a standard chow diet (LabDiet^®^ Rodent Diet 5001 contains as a percentage of total calories: 28% protein, 60% carbohydrate, and 12% fat) and water in the manner of ad-libitum up to 8 weeks aged. To remove endogenous androgens, mice were orchiectomized bilaterally at age of 8 weeks. After the operation, we provided a high-fat diet (a percentage of total calories: 17.3% protein, 38.4% carbohydrate, and 43.4% fat) and injected testosterone dissolved in corn oil, 2.5 mg/mL every 3 days during 4 weeks, from recovery period for 2 weeks. After6 weeks from the operation, the mice were sacrificed. The number of mice used for the experiment was 4, 4, 4, 5, and 4 for each group: Normal diet-Navie (ND-Naive), Normal diet-Orchiectomized (ND-ODX), High fat diet-Navie (HD-Naive), High fat diet-Orchiectomized (HD-ODX), and High fat diet-ODX-treated testosterone (HD-ODXT).

### 4.3. Blood Alanine Transferase Level

Serum was diluted by 1/5 fold with sterilized phosphate-buffered saline. Plasma ALT levels were measured with FUJI DRI-CHEM SLIDE (ALT-3250) by DRI-CHEM4000 (Fuji Film, Minato, Tokyo, Japan). 

### 4.4. Hematoxylin & Eosin Staining and Masson’s Trichrome Staining

A portion of mice livers was fixed in 10% buffered formaldehyde solution and embedded in paraffin. The paraffin samples were sectioned to 4 µm and attached on silane coated slide. Following de-waxed and re-hydrated, the samples were stained with hematoxylin and eosin (H&E) or Masson’s trichrome (BioGnost, MST-100T, BioGnost Ltd., Medjugorskae, Zagreb, Croatia). The stained samples were examined using VM600 Digital Slide Scanning System (Motic, Schertz, TX, USA). 

### 4.5. Oil-Red-O Staining

Frozen liver tissues were sectioned by 7 µm after being embedded with OCT compound and attached to silane coated slide. Following drying for 10 min in RT, slides were fixed with formalin for 20 min. Slides were washed with tap water for 10 min and subsequently rinsed with 60% isopropanol. Staining was processed with Oil-Red-O solution (3 g/L) for 15 min. After washing with 60% isopropanol, slides were stained with hematoxylin for 30 s and washed with distilled water. The stained samples were examined using a light microscope after being mounted in an aqueous mounting medium.

### 4.6. Cell Culture

To reveal the relationship between a level of AR expression and LKB1, two types of cell lines were used in this study. One is the SNU-423 cell line acquired from the Korean Cell Line Bank (KCLB, 00423) and the other is Hep3B (KCLB, 88064). Unlike the normal property of liver cells including Hep3B, SNU-423 possesses a significant level of AR expression. 

Both cell lines were grown at 37 °C in a 5% CO_2_ atmosphere. SNU-423 was cultured by RPMI 1640 (LM 011-01, Welgene, Gyeongsan, Gyeongsangbuk, Korea) with 5% fetal bovine serum (FBS) and 1% penicillin-streptomycin (P/S) and Hep3B by Dulbecco’s modified Eagle’s medium(DMEM)-High glucose (LM 001-05, Welgene) with 5% FBS and 1% P/S. Additionally, a starvation solution which is DMEM/F-12 (LM 002-05, Welgene) with 2% charcoal dextran fetal bovine serum (CD-FBS) and 1% P/S, was used to deplete endogenous steroid hormones.

Fatty acids (Palmitic acid 330 μM, Oleic acid 670 μM), dissolved in absolute ethanol (K46253383 506, EMD Millipore Corp, Burlington, MA, USA), were treated and incubated for 24 h, to provide a similar condition to the mice fed high-fat diet. Testosterone (T0027, Tokyo chemical industry CO. LTD, Chuo, Tokyo, Japan) and bicalutamide (B3206, Tokyo chemical industry CO. LTD), dissolved in dimethyl sulfoxide (DMSO) (#MKCH9998, Sigma-Aldrich, St. Louis, MO, USA), were supplemented with concentration 10 nM/L and 5 uM/L [30] respectively, as per one well of six-well tissue culture plates and incubated for each indicated hours.

### 4.7. Gene Overexpression Assay

Hep3B cells were seeded into six-well tissue culture plates and cultured until they take possession of space of 70%. Transient transfection of *AR* expression vectors (2.5 µg), a pSG5 eukaryotic expression vector encoding the human *AR* [31], extracted with NucleoBondXtra Midi kit (740,410.10, MACHEREY-NAGEL GmbH & Co. KG, Neummann Neander Str, Düren, Germany), were performed in 1.5 mL Opti-MEM (31,985,070, Invitrogen, Waltham, MA, USA) containing lipofectamine2000 (11668019, Invitrogen) for 18 h. After the transfection, the cells were cultured with 2 mL of starvation solution for 48 h. The cells were washed with PBS twice and harvested through scraping. Following centrifugation, cell pellets were processed for being used for Western blotting or total RNA extraction.

### 4.8. Gene Knock-Down Assay

SNU-423 cells were seeded into six-well tissue culture plates and cultured until they take possession of space of 70%. *LKB1* specific siRNA consisted of (5′-GUACUUCUGUCAGCUGAUUdTdT-3′ and 5′-AAUCAGCUGACAGAAGUACdTdT-3′) [32] and was purchased from Bionics Inc. (Seongdong, Seoul, Korea). Transfection were performed in 1.5 mL Opti-MEM (31985070, Invitrogen) containing lipofectamine2000 (11668019, Invitrogen) for 18 h. After the transfection, the cells were cultured with 2 mL of RPMI 1640 containing with/without fatty acids (Palmitic acid 330 μM, Oleic acid 670 μM) for 24 h. The cells were washed with PBS twice and harvested through scraping. Following centrifugation, cell pellets were processed for being used for Western blotting.

### 4.9. Total RNA Extraction and Real-Time Quantitative PCR

Total RNA was extracted using TRIzol Reagent (15596-026, Life technologies, Carlsbad, CA, USA) in both liver and cell samples. Reverse transcription was performed with 1.5 µg of total RNA and Reverse transcriptase kit (SG-cDNAS100, Smartgene, Ecublens, Lausanne, Switzerland) following the manufacturer’s protocol. Quantitative PCR (real-time PCR) was executed using Excel Taq Q-PCR Master Mix (SG-SYBR-500, Smartgene) and StratageneMx3000P (Agilent Technologies, Santa Clara, CA, USA). Primers used in real-time PCR were manufactured by Bionics Inc. (Seoul, Korea) or Genotech (Yuseong, Daejeon, Republic of Korea). *mRPLP0* and *18 s* were used as a control in in vivo samples and in vitro respectively. The primers used for real-time PCR are in Table 2. All experiments were run more than triplicate, and mRNA values were calculated based on the cycle threshold and monitored for an amplification curve.

### 4.10. Western Blotting

Both protein samples of livers and the cells of SNU-423 and Hep3B were extracted by using protein lysis buffer, called T-PER reagent (78510, Thermo Fisher Scientific, Waltham, MA, USA) and quantified by Bradford assay with PRO-Measure solution (#21011, iNtRON Biotechnology, Seongnam, Gyeonggi, Korea). The samples were run SDS-PAGE electrophoresis on 10% polyacrylamide gels and transferred to the membrane. And the membranes were blocked with 30 mg/mL BSA100 (9048-46-8, LPS solution), diluted TBS-T buffer (04870517TBST4021, LPS solution, Daeduk, Daejeon, Korea). Primary antibodies (Table 1) were operated overnight at 4 °C. Following the step, the membranes were washed with TBS-T and secondary antibodies (Table 1) were operated identically. Results were detected with ECL solution (XLS025-0000, Cyanogen, Via degli Stradelli Guelfi, Bologna, Italy) and Chemi Doc (Fusion Solo, VilberLourmat, Lamirault, Collégien, France).

### 4.11. Chromatin Immunoprecipitation Assay

To clarify whether androgen receptor binds with genes related to de novo lipogenesis, we performed chromatin immunoprecipitation (ChIP). The sample was prepared with SNU-423 cells, which were supplemented with testosterone (10 nM) and incubated for 2 h. DNA-protein cross-linking was achieved by using formaldehyde (1% final) at room temperature for 10 min. The cells were washed with cold PBS twice and resuspended by lysis buffer (50 mM Tris–HCl, pH 8.1, 10 mM EDTA, 1% SDS). Then, the sample was sonicated on ice at power 25 for 10 s pulses 10 times, using a VirSonic 100 (Virtis) sonicator. After the sonicated sample was centrifuged at 13,000 rpm for 15 min at 4 °C, chromatin (84.56 µg) was diluted in 2.5× ChIP dilution buffer (0.5% Triton X-100, 2 mM EDTA, 100 mM NaCl, 20 mM Tris–HCl, pH 8.1) and incubated overnight at 65 °C with anti-AR monoclonal antibody (sc-7305; Santa Cruz Biotechnology, Dallas, TX, USA) or mouse IgG antibody as a control (sc-2025; Santa Cruz Biotechnology) and 50 μL of Dynabeads Protein A (10003D; Invitrogen).The beads were washed 6 times with LiCl buffer (1% NP-40, 500 mM LiCl, 1% Na-deoxycholate, pH 8.0, 100 mM Tris–HCl, pH 8.1), and washed briefly with TE buffer (10 mM Tris–HCl, pH 7.5, 1 mM EDTA, pH 8.0), in turn. Then, the washed sample was de-crosslinked (1% SDS, 0.1 M NaHCO_3_) at 65 °C overnight and was purified using the G-Spin DNA Extraction Kit (17045; iNtRONBiotechology). AR standard ChIP enrichments were quantified by PCR analysis using specific primers.

### 4.12. Statistical Analysis

Data are reported as the mean ± standard deviation (SD). Differences between means were obtained by Student’s *t*-test and one-way ANOVA followed by a Dunnett post analysis was performed using Graph Pad Software (GraphPad Inc., San Diego, CA, USA).

## 5. Conclusions

Our data all suggest that NAFLD development was accentuated in conditions where the supply of androgens is limited. When AMPK-ACC signaling is generally considered the mainstream in de novo lipogenesis, testosterone modulates the signaling. Intuitively, testosterone could lead to the reduction of hepatic TG contents. We confirmed that LKB1 regulated AMPK-ACC signaling and that testosterone interacts directly with the LKB1 gene to evoke a direct AR binding. This study has increased our understanding of how testosterone acts to regulate NAFLD development and its relevance to androgen-responsive LKB1 in the male liver.

## Figures and Tables

**Figure 1 ijms-22-07904-f001:**
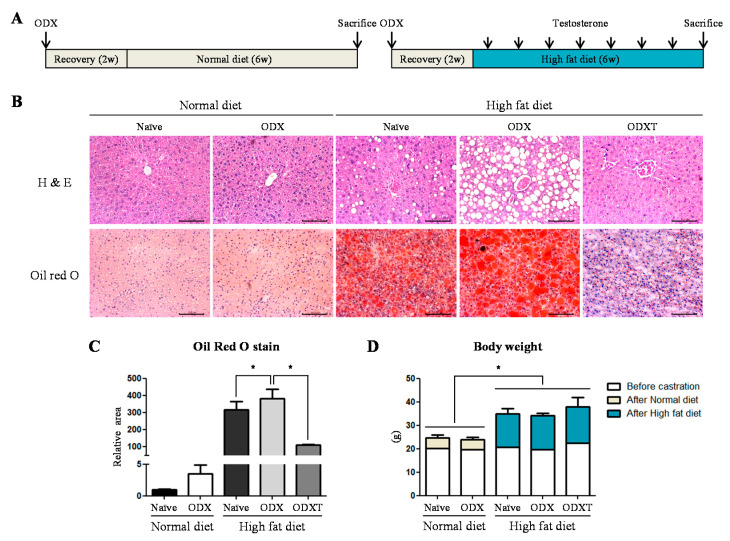
Depending on high energy diet and androgen, the lipid accumulation level within the liver fluctuates in male mice. (**A**) A schematic diagram shows the schedule of an animal experiment. After surgical castration (ODX) to deprive male mice of endogenous androgen, the mice were fed with a high-fat diet (HD) and injected with testosterone (T) for 6 weeks. (**B**) Their livers were used to hematoxylin and eosin stain (H&E) and Oil-Red O stain, to examine the degree of accumulated lipid. Scale bar = 100 µm. (**C**) In the Oil-Red O stain, the areas of lipid droplets were quantified by image J program. (**D**) To consider whether hepatic TG accumulation influences bodyweight, the weight was measured. The values stand for means +/− S.D. * *p* < 0.05 was compared to groups indicated.

**Figure 2 ijms-22-07904-f002:**
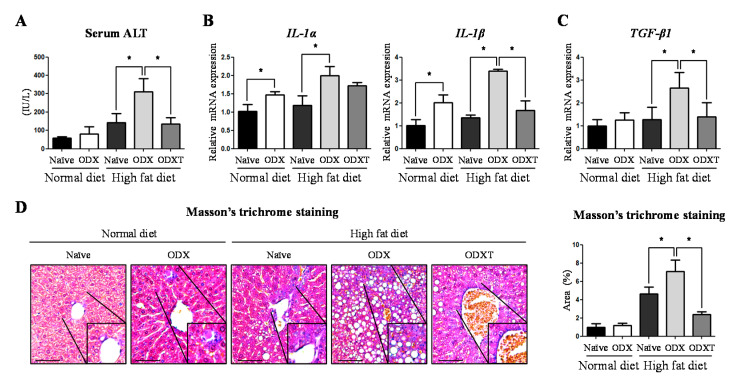
The hepatic steatosis depending on high energy diet and androgens is progressed to steatosis-induced NASH with fibrosis. (**A**) Serum alanine aminotransferase (ALT) was measured as the damage marker of hepatocytes. (**B**) To consider the influence on the inflammatory response, *interleukin-1α* (*IL-1α*) and *IL-1β* mRNA levels were determined by quantitative RT-PCR, using *mouse 60S acidic ribosomal protein P0* (*mRPLP0*) as an internal control. (**C**) As a marker of fibrosis, *transforming growth factor-β* (*TGF-β*) mRNA levels were determined by quantitative RT-PCR, using *mRPLP0* as an internal control. (**D**) To show the development of fibrosis in livers, Masson’s trichrome staining was performed and the fibrotic areas were quantified by image J program. Scale bar = 100 µm. The values stand for means +/− S.D. * *p* < 0.05 was compared to groups indicated.

**Figure 3 ijms-22-07904-f003:**
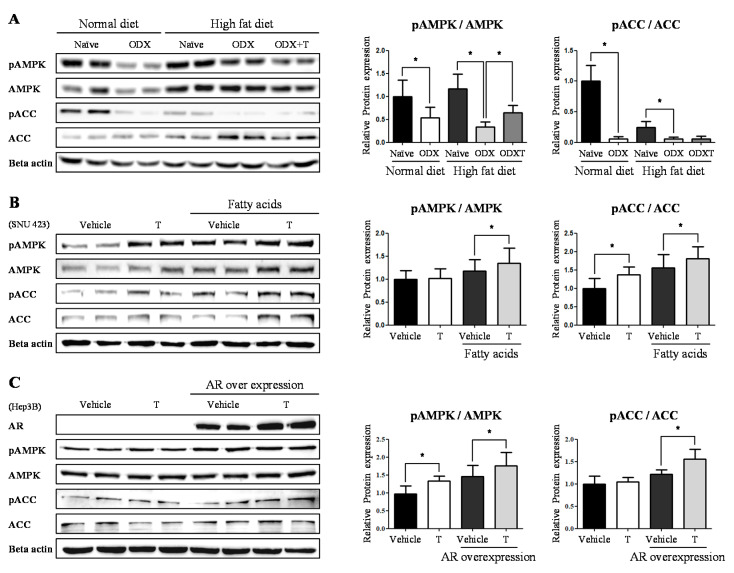
When androgens are sufficient, AMPK-ACC signaling is phosphorylated. (**A**) The protein level was determined by Western blotting of male mice livers. The quantification was arranged as the ratio of phosphorylated form and total form, in AMPK and ACC. Beta-actin (Rabbit) was used as an internal control. (**B**) The proteins level was determined by Western blotting, with SNU-423 cells supplemented with fatty acids (Palmitate 330 μM, Oleate 660 μM). Testosterone (T) (10 nM) was treated and incubated for 6 h. The quantification was arranged as the ratio of phosphorylated form and total form, in AMPK and ACC. Beta-actin (Rabbit) was used as an internal control. (**C**) The proteins level was determined by Western blotting, with Hep3B cells after overexpressed to *AR*. Testosterone (T) (10 nM) was treated and incubated for 6 h. The quantification was arranged as the ratio of phosphorylated form and total form, in AMPK and ACC. Beta-actin (Rabbit) was used as an internal control. The values stand for means +/− S.D. * *p* < 0.05 was compared to groups indicated.

**Figure 4 ijms-22-07904-f004:**
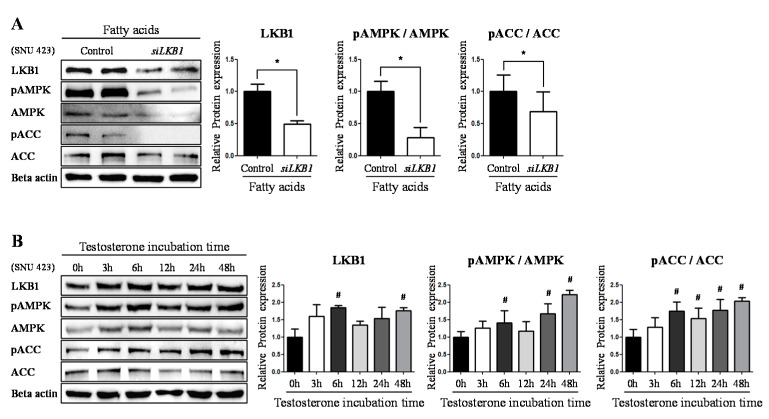
LKB1 influences phosphorylation of AMPK-ACC signaling and was influenced by testosterone. (**A**) *LKB1* knocked down assay by *LKB1* specific siRNA (*siLKB1*) was performed with SNU-423 cells supplemented with fatty acids (Palmitate 330 μM, Oleate 660 μM). The proteins level was determined by Western blotting. The levels of LKB1 expression and phosphorylation of AMPK and ACC was quantified and beta-actin (Mouse) was used as an internal control. (**B**) Testosterone time course assay was performed with SNU-423 cells supplemented with testosterone (10 nM). The proteins level was determined by Western blotting. The levels of LKB1 expression and phosphorylation of AMPK and ACC was quantified and beta-actin (Rabbit) was used as an internal control. The values stand for means +/− S.D. * *p* < 0.05 was compared to groups indicated. # *p* < 0.05 was compared to 0 h, which is testosterone incubation time.

**Figure 5 ijms-22-07904-f005:**
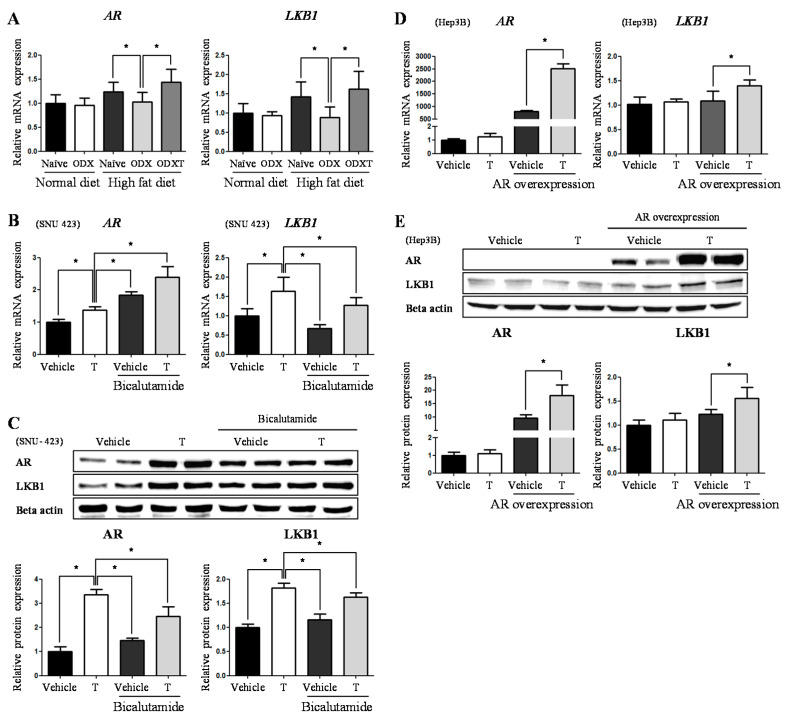
Testosterone increases the AR and LKB1 expression. Cell samples were treated with testosterone (T) (10 nM) with/without bicalutamide (5 μM), incubated for 6 h after the treatment. (**A**) Using male mice liver, androgen receptor (*AR*) and *LKB1* mRNA levels were determined by quantitative RT-PCR, using mRPLP0 as an internal control. (**B**) In vitro experiment using SNU-423 cells, *AR* and *LKB1* mRNA level was determined by quantitative RT-PCR, using *18s* as an internal control. (**C**) In vitro experiment using SNU-423 cells, AR and LKB1 proteins level is determined by Western blotting. Beta-actin (Rabbit) was used as an internal control. (**D**) In *AR* overexpression assay using Hep3B cells, *AR* and *LKB1* mRNA level was determined by quantitative RT-PCR, using 18s as an internal control. (**E**) In *AR* overexpression assay using Hep3B cells, AR and LKB1 proteins level is determined by Western blotting. Beta-actin (Rabbit) was used as an internal control. The values stand for means +/− S.D. * *p* < 0.05 was compared to groups indicated.

**Figure 6 ijms-22-07904-f006:**
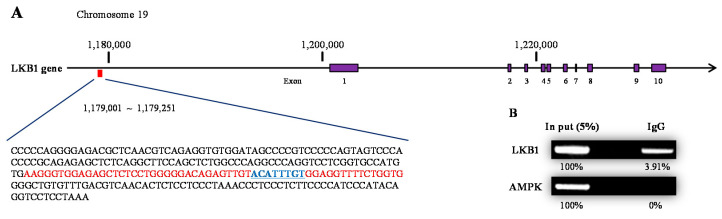
Testosterone cooperating with AR is recruited in the promotor of the *LKB1* gene. In SNU-423 cells (human), testosterone (10 nM) was added to the existing media, and the cells were incubated for 2 h. (**A**) The region of AR occupancy on the *LKB1* gene is located 26,931 bp upstream of the transcription initiation site (1,205,778 in chromosome 19). Probe sequence and ARE motif were indicated with red and blue, respectively. (**B**) Binding with *LKB1* and *AMPK* gene promoters were detected by antibodies of IgG and AR. The binding intensity was quantified comparing with the ‘5% input’ sample.

**Table 1 ijms-22-07904-t001:** Antibodies used for Western Blotting and Chromatin Immunoprecipitation Assay.

**Primary Antibodies**	**Type**	**Lot.**	**Inc.**
Beta-actin	Mouse monoclonal	sc-47778	Santa Cruz biotechnology
Beta-actin	Rabbit polyclonal	sc-130656	Santa Cruz biotechnology
ACC	Rabbit monoclonal	#3676T	Cell signaling technology
pACC	Rabbit monoclonal	#11818T	Cell signaling technology
AMPK-alpha	Rabbit monoclonal	#5831T	Cell signaling technology
pAMPK-alpha	Rabbit monoclonal	#2535T	Cell signaling technology
AR	Rabbit monoclonal	#5153	Cell signaling technology
LKB1	Rabbit polyclonal	#A2122	Company ABclonal, Inc.
**Secondary Antibody**	**Type**	**Lot.**	**Inc.**
Anti-Mouse IgG	Goat	121507	Jackonimmuno
Anti-Rabbit IgG	Mouse	123213	Jackonimmuno

**Table 2 ijms-22-07904-t002:** Primers used for Real-Time Quantitative PCR.

Gene Name	Upper Primer (5′-3′)	Lower Primer (5′-3′)	Species
*18 s*	GGA CAC GGA CAG GAT TGA CA	AGA CTG TGT CCC TGT GGA GA	Human

*AR*	GAC GAC CAG ATG GCT GTC ATT	GGG CGA AGT AGA GCA TCC T	Human

*LKB1*	GGT TCC GGA AGA AAC ATC CT	TGT GAC TGG CCT CCT CTT CT	Human

*RPLP0*	GCA GCA GAT CCG CAT GTC GCT CCG	GAG CTG GCA CAG TGA CCTCAC ACG G	Mouse

*AR*	CTG GGA AGG GTC TAC CCAC	GGT GCT ATG TTA GCG GCC TC	Mouse

*LKB1*	TAT GTG GCA TGC AGG AGA TG	TGG TGG TGA GTA GCA GGT TG	Mouse

*IL-1* *α* *-*	AGT ATC AGC AAC GTC AAG CAA	TCC AGA TCA TGG GTT ATG GAC TG	Mouse

*IL-1β*	GAA ATG CCA CCT TTT GAC AG	CTG GAT GCT CTC TCA TCA GGA CA	Mouse

*TGF-* *β1*	GAC GTC ACT GGA GTT GTA CG	GGT TCA TGTCAT GGA TGG TG	Mouse


## Data Availability

The data presented in this study are available on request from the corresponding author.

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
