# Peer review of "Hepatic LKB1 Reduces the Progression of Non-Alcoholic Fatty Liver Disease via Genomic Androgen Receptor Signaling"

_ijms, 2021, doi:10.3390/ijms22157904_

Round 1
Reviewer 1 Report
The present manuscript addresses the regulation of hepatic LKB1 by testosterone in the non-alcoholic fatty liver. The study only verified that the regulation of LKB1 by testosterone already described in adipocytes also occurs in hepatocytes. Furthermore, the effect of LKB1 on the regulation of liver metabolism has also been recently described. Therefore, the novelty of the study is scarce. However, the experiments are conducted in a rational and adequate way, although some questions must be addressed to reach more solid conclusions.
1) Are the authors sure that their high-fat diet model produces NAFLD? It is striking that with six weeks of a high-fat diet, the weight of the animals does not increase. Neither p-AMPK nor LKB1 levels increase. Is it possible that this pathway is not actually active in the NAFLD (high fat diet) model? If so, is this due to testosterone levels? Hepatic testosterone levels should be measured to try to rationalize the true role of LKB1 in the progression of NAFLD.
2) The changes in the expression levels of AR and LKB1 in OVDX animals are very scarce. The levels of these proteins should be measured by western blot in all experimental groups.
3) This reviewer doubts whether SNU-423 cells (tumor cells with a high level of androgen receptors) are the most suitable for assessing the physiological dependence of LKB1 on testosterone levels. Although the use of these cells is excellent in contrast to HepG3 (this reviewer values ​​this experiment very positively) it is evident that the effect of testosterone on these pathways in isolated hepatocytes has to be compared.
Minor comments:
1) Histology and western blot images are of poor quality. This needs to be improved in order to get changes appreciated.
2) The text in the figures is too small and illegible.
3) "Intuitively, testosterone leads to the reduction of TG contents in both the liver and hepatocyte." This is not a conclusion.
Author Response
We have made the following editorial changes in response to the reviewer’s comments. Please see the attached file.
Sincerely,
Eui-Ju Hong

Reviewer 2 Report
Comments for authors
- Accumulation of lipid in the liver can be traced by increased uptake of fatty acids into the liver, impaired fatty acid beta oxidation, or the increased incidence of de novo In the present study, author focused on the de novo lipogenesis under condition of androgen. Are there any evidences about beta-oxidation in androgen treated mice or cells?
- To reconstruct the androgen influence of AMPK-ACC signaling in vivo, the authors used SNU-423 cells expressing AR. The author introduces the AR into Hep3B cell in which it was a low level of AR, and then treated androgen and observed the LKB1 expression. I hope you add the information about AMPK-ACC signaling.
- There is a missing bar scale for Figure 1B.
- There is a missing magnification and bar scale for Figure 2D.
- The author should check Figure 5E because labels were mismatched and/or lack.
- Pease provide the information of AR expression vector information.
- Please scale up the font size of the Figures.
Author Response

(The authors gave the same response as above.)

Round 2
Reviewer 1 Report
Most of my concerns have been resolved.
Reviewer 2 Report
All raised issues are properly addressed.